# Antimicrobial, Antibiofilm and Toxicological Assessment of Propolis

**DOI:** 10.3390/antibiotics12020347

**Published:** 2023-02-08

**Authors:** Maria Cristina Queiroga, Marta Laranjo, Nara Andrade, Mariana Marques, Ana Rodrigues Costa, Célia Maria Antunes

**Affiliations:** 1MED–Mediterranean Institute for Agriculture, Environment and Development & CHANGE–Global Change and Sustainability Institute, Institute for Advanced Studies and Research, Universidade de Évora, Pólo da Mitra, Ap. 94, 7006-554 Évora, Portugal; 2Departamento de Medicina Veterinária, Escola de Ciências e Tecnologia, Universidade de Évora, Pólo da Mitra, Ap. 94, 7006-554 Évora, Portugal; 3Universidade Paulista, Campus Petrolina-PE, Av. Barão do Rio Branco, 700-862 - Centro, Petrolina - PE, CEP: 56304-260, Brazil; 4Instituto de Ciências da Terra, ICT, Universidade de Évora, 7006-554 Évora, Portugal; 5Departamento de Ciências Médicas e da Saúde, Escola de Saúde e Desenvolvimento Humano, Universidade de Évora, 7006-554 Évora, Portugal; 6Centro Académico Clínico do Alentejo, C-TRAIL, 7000-671 Évora, Portugal

**Keywords:** propolis, antimicrobial, antibiofilm, toxicological, *Staphylococcus*, fibroblast, keratinocyte

## Abstract

Antimicrobial resistance is a serious problem for the control of infections and infectious diseases. Propolis is a substance produced by honeybees with antimicrobial and antibiofilm properties. To consider propolis as an alternative to the use of antimicrobials for infection control, we assessed its antimicrobial and antibiofilm activities. To assess propolis for topical medical use, toxicological studies were also performed. A Portuguese 70% propolis ethanolic extract was chemically evaluated and studied for antimicrobial activity on staphylococcal field isolates (n = 137) and antibiofilm action (n = 45). Cell toxicological assessment was performed using keratinocytes and fibroblasts. Pinobanksin, chrysin, acacetin, apigenin, pinocembrin, and kaempferol-dimethyl-ether were detected. All 137 isolates were susceptible to 6.68 mg/mL or lower propolis concentration (80% isolates were susceptible to <1 mg/mL). The mean percentage of biofilm inhibition was 71%, and biofilm disruption was 88.5%. Propolis (<1 mg/mL) was well-tolerated by fibroblasts and moderately tolerated by keratinocytes. The combined antimicrobial and antibiofilm effect of propolis, together with its low toxicity to connective tissue and epithelial cells, suggests a good applicability for topical antibacterial treatment. Therefore, propolis seems to be a good alternative to antimicrobials for the treatment of infections with *Staphylococcus* spp. that deserves to be evaluated in vivo.

## 1. Introduction

Antimicrobial resistance (AMR) is a highly concerning issue, which has driven a lot of scientists to study alternative molecules and products to be used in the control of infections and infectious diseases. Furthermore, from a One Health perspective, antimicrobial resistance genes can be laterally transferred between bacteria that can be transmitted between animals and humans or vice versa [1,2].

Another concerning issue refers to biofilm-producing bacteria, which are responsible for the majority of infections [3]. Biofilms are bacterial consortia embedded in a matrix of polysaccharides, protein, and environment DNA [4], which protect bacteria from immune response mechanisms and pharmaceutical activity, hampering infection resolution [5]. Thus, investigation on new antimicrobial molecules/substances that are economically and ecologically sustainable is utterly relevant.

As an alternative to current antimicrobials, propolis deserves to be further studied. Propolis is a mass produced by honeybees *Apis mellifera*, which they use to protect the honeycomb against undesirable visitors and to maintain the right temperature [6]. To produce this substance, the bees use their own salivary secretion together with resins harvested in different plants [6]. Propolis is rich in flavonoids and other phenolic compounds and may also contain terpenoids [7]. These plant secondary metabolites are important to protect the plants from herbivores and microbial infection and are also potential sources for new natural drugs, antimicrobials, insecticides, and herbicides [8].

Additionally, propolis has been used for a long time in the preparation of different medicines to treat several pathologies and is now quite popular in Europe due to its antibacterial activity [9,10,11]. Its antibacterial mode of action was assigned to physicochemical changes in the bacterial cell wall, resulting from the interaction with the different constituents of propolis, increasing surface permeability, through the formation of intramolecular hydrogen bonds and subsequent hydrophobic interactions between the propolis ethanolic extract (PEE) antimicrobial compounds and the cell wall or cytoplasmic membrane [12]. Moreover, propolis activity against bacterial biofilms has been reported [11,13,14,15].

In the last years, we have been studying propolis ethanolic extracts for the use in ruminant mastitis control with promising results [16]. Often, intramammary infections in animals are caused by *Staphylococcus aureus* and coagulase negative staphylococci (CNS) [17]. Such bacteria are also responsible for other animal and human pathologies [18] and frequently show AMR and the ability to produce biofilms [19].

Besides their antibacterial and antibiofilm activities, propolis extracts have shown differential effect on the viability of several carcinoma and sarcoma cell lines being better tolerated by the latter [20,21,22]. The effect of propolis extracts on primary cell line viability, which are better models for normal epithelial and connective tissues, is less characterized; nevertheless, one study shows that the extracts are well-tolerated by a fibroblast lineage [20] and has shown to promote keratinocyte proliferation and migration [23], thus suggesting that it might be well-tolerated by animal tissues.

To consider propolis as an alternative to the use of classical antimicrobials for infection control, we assessed propolis ethanolic extract antimicrobial activity through both its inhibitory activity on biofilm formation and its ability to eliminate established biofilm. The toxicological activity was evaluated on the mammalian cell cultures of fibroblasts and keratinocytes, as models for connective and epithelial tissues, respectively, for use in human and veterinary medicine.

## 2. Results

### 2.1. Propolis Major Chemical Groups and Chemical Profile

The chemical composition of the propolis ethanolic extract was as follows: the total phenolics content was 67.6 ± 2.8 mg GAE/g, the flavonoids content was 54.8 ± 1.7 mg QE/g, anthocyanins were not detected, and the tannins content was 36.9 ± 0.4 mg ECC/g.

The chemical profile as defined by UPLC-QTOF-MS/MS detected the presence of pinobanksin, chrysin, acacetin, apigenin, pinocembrin, and kaempferol-dimethyl-ether (Table 1). The different compounds were identified based on previous studies [24,25,26,27,28,29].

### 2.2. Antimicrobial Properties of Propolis

All 137 staphylococci isolates revealed susceptibility to 6.68 mg/mL or lower concentration of this propolis ethanolic extract.

The number of isolates susceptible to different propolis ethanolic extract concentrations is shown in Figure 1.

The susceptibility of the different *Staphylococcus* species (including field isolates and control strains) to the propolis ethanolic extract is given in Table 2.

There are significant differences between *Staphylococcus* species regarding their susceptibility to the propolis ethanolic extract (F = 31.88, *p* < 0.001). *S. hyicus* is the most susceptible, whereas *S. warneri* is the less susceptible species.

Regarding the most important staphylococcus human pathogens, *S. aureus* and *S. epidermidis*, the susceptibility to the propolis ethanolic extract is 0.82 mg/mL (±0.96) and 0.48 mg/mL (±0.55), respectively.

### 2.3. Antibiofilm Properties of Propolis

Both biofilm formation inhibition and established biofilm disruption activities have been evaluated in terms of the number and percentage of affected isolates, as well as the percentage of biofilm reduction (inhibition or disruption).

The mean propolis ethanolic extract ability for inhibiting biofilm formation is 71% (Table 3).

Considering the biofilm disruption, the mean activity was 88.5% (Table 4).

This brown propolis extract was able to inhibit the formation of biofilm in 34 out of 45 staphylococci. Regarding the established biofilm disruption, the propolis ethanolic extract was effective on 36 isolates.

### 2.4. Toxicological Studies

Compared with the control, propolis (0.025–2.5 mg/mL) induced the loss of keratinocyte viability in a dose–response manner, showing a cell viability decline of 20% to 100%, following the increase in propolis ethanolic extract concentration. On the contrary, fibroblasts’ viability was not affected by propolis in the concentration range of 0.025–0.25 mg/mL. Viability decreased to 40% at 1 mg/mL but was significantly diminished by the highest concentration (2.5 mg/mL) (Figure 2).

## 3. Discussion

The final goal of the present study is to contribute toward the reduction of antimicrobial usage. Each time antimicrobials are used for the control of both animal and human infections; a selection pressure is applied over resistant bacterial strains. According to the predictive statistical models of Murray et al. [30], antimicrobial resistance was estimated to be directly responsible for 1.27 million deaths in 2019. Moreover, the same report estimated that 4.95 million deaths were further associated with bacterial AMR in 2019 [30].

“Preventing Antimicrobial Resistance Together” was the theme of the 2022 World Antimicrobial Awareness Week (WAAW) defined by the quadripartite organizations, namely the Food and Agriculture Organization of the United Nations (FAO), the United Nations Environment Programme (UNEP), the World Health Organization (WHO), and the World Organisation for Animal Health (WOAH, founded as OIE) (https://www.who.int/news/item/04-07-2022-world-antimicrobial-awareness-week-2022-preventing-antimicrobial-resistance-together (accessed on 27 December 2022)).

In both human and veterinary medicine, efforts must be combined to severely reduce the use of antimicrobials. Prevention of infections through access to vaccines, sanitation, and hygiene based on good practices in food and agriculture production, and sound management of waste and wastewater, are measures specifically indicated by the abovementioned quadripartite.

With our study, we intend to go a step further and propose alternative compounds to antimicrobials, such as propolis.

Compared with other propolis, the contents in total phenolics, flavonoids, and tannins in the studied propolis ethanolic extract are much lower, which has been reported to be associated with a higher bactericidal activity [16,31]. However, other works described the opposite effect [32,33]. Additionally, different authors did not observe a direct correlation between the minimal inhibitory concentration (MIC) and the total content in flavonoids and other phenolic compounds [34,35].

Concerning the antibiofilm effect of propolis, total phenolics, flavonoids, and condensed tannins have been previously associated with the inhibition of biofilm formation ability [14,15,16]. On the other hand, a positive correlation has been reported between condensed tannins and biofilm disruption ability, while flavonoids and other total phenolics were negatively correlated with this ability [16].

Regarding the chemical profile of this propolis ethanolic extract, assessed by UPLC-QTOF MSE experiments, pinobanksin, chrysin, acacetin, apigenin, pinocembrin, and kaempferol-dimethyl-ether were detected. These six flavonoids were earlier linked to a positive influence on the bactericidal activity of propolis [16]. Moreover, the antimicrobial activity of pinocembrin [36,37], apigenin, chrysin, and kaempferol [37] has also been reported by other authors.

The studied propolis ethanolic extract has inhibited the biofilm formation ability and promoted the biofilm disruption capacity. Other studies have also reported the inhibition of biofilm formation in *S. aureus* [15], *S. epidermidis* [11], and *Streptococcus mutans* [14]. Nevertheless, different flavonoids may have distinct antibiofilm effects [38]. In a previous study, apigenin, pinocembrin, and kaempferol-dimethyl-ether showed a positive effect on the biofilm disruption ability while not influencing the inhibition of biofilm formation [16]. On the other hand, pinobanksin, chrysin, and acacetin decrease the ability of inhibiting biofilm formation, although they are not interfering with the disruption ability [16].

The toxicological studies revealed that propolis was well-tolerated by dermal fibroblasts and moderately tolerated by epithelial keratinocytes in the range between 0.025 and 1.0 mg/mL. According to the OECD guidelines, a cell viability of 40% is considered good, while a 20% cell viability can be considered as moderate in chronic exposures [39,40], where cell viabilities over 15% are considered acceptable. Moreover, in the current study, cell viability was evaluated in vitro, with the cells completely unprotected, where cell membranes are directly exposed to propolis. In vivo, cells are part of tissues with the natural protection of an extracellular matrix; thus, deleterious effects are expected to be lower. Furthermore, other studies performed in tumor epithelial and fibroblast cell lines have shown dose–response cytotoxicity with a half-maximal inhibitory concentration (IC50) of 0.04–0.16 mg/mL [20,21,22]. Additionally, propolis has shown healing properties by inducing the proliferation and migration of primary lineages of keratinocytes [23]. Therefore, we may suggest that this propolis ethanolic extract can be used without significant damage to fibroblasts and keratinocytes in concentrations below 1 mg/mL. Furthermore, due to its beneficial properties for epithelia at a low concentration, propolis is a good candidate to be used for the healing process, during the recovery phase, after infection has recessed.

A propolis ethanolic extract concentration of 1 mg/mL is bactericidal for most staphylococcal isolates studied, namely *S. aureus* and *S. epidermidis*. The same dose would be efficient for antibiofilm purposes as the experiments were performed with half the minimum bactericidal concentration (MBC) of the propolis ethanolic extract. Half MBC of the propolis ethanolic extract was able to inhibit the formation of biofilm on 80.7% isolates, reducing the biofilm in 71%. For established biofilm disruption, half MBC was effective in 82.9% isolates, reducing the biofilm in 88.5%.

The in vitro experiments highly suggest that propolis is a good alternative to classical antimicrobials that deserves to be further studied.

## 4. Materials and Methods

### 4.1. Propolis Collection and Extract Preparation

A brown propolis sample was collected in an apiary near Monchique, district of Faro in Algarve, Portugal (latitude 37,2859 and longitude -8,55594). The vegetation around the apiary, according to the beekeepers, mainly consists of *Cistus ladanifer*, *Arbutus unedo*, *Lavandula stoechas*, *Thymus serpyllum*, and *Eucalyptus* sp.

Propolis ethanolic extract was prepared according to the official standards for extract production in Brazil [16]. Briefly, a 30% propolis ethanolic extract was prepared by cold maceration of 300 g of raw propolis in 700 mL of 70° ethanol and was kept at room temperature, protected from light, for 45 days. The extracts were then filtered through a sterile filter paper and kept refrigerated at 4 °C, in amber bottles, until use.

### 4.2. Propolis Major Chemical Groups and Chemical Profile

The propolis ethanolic extract content in total phenolics, flavonoids, condensed tannins, and anthocyanins was determined by spectrophotometric methods as described before [16]. Total phenolics were determined by the Folin-Ciocalteau method [41]. Flavonoids were determined by spectrophotometry and the content in condensed tannins by colorimetry [42]. The anthocyanin content was measured by the differential pH method [43].

Chromatographic methods were used to identify some individual compounds from the propolis ethanolic extract as previously described [16]. The propolis ethanolic extract phenolic profile was recorded at 290 nm, and the compounds were tentatively identified by ultrahigh-pressure liquid chromatography along with quadrupole time-of-flight mass spectrometry (UPLC-QTOF-MS/MS) (ThermoFisher Scientific, Waltham, MA, USA) as flavonoids (flavonol/flavone, isoflavone, flavanone, and chalcones), non-flavonoids, and triterpenes.

### 4.3. Bacterial Isolates

Bacterial isolates were obtained from aseptically collected sheep and goat mastitic milk samples and were identified to the species level using API-Staph (Biomérieux, Marcy-l’Étoile, France) or Vitek 2 Compact (Biomérieux, Marcy-l’Étoile, France) [16].

To assess the antimicrobial activity of a propolis ethanolic extract, 137 staphylococci isolates (35 *Staphylococcus aureus* and 102 CNS) were used. Seven reference strains, five *S. aureus* (ATCC 25923, ATCC 29213, COL, FRI 472, and FRI 913) and two *S. epidermidis* (ATCC 12228 and ATCC 35984), served as controls.

Biofilm production was evaluated as described by Laranjo et al. [44], following Merino et al. [45] with some modifications. Biofilm-producing staphylococci isolates comprised 26 *S. aureus*, 7 *S. chromogenes*, 4 *S. warneri*, 3 *S. auricularis*, 2 *S. simulans*, 1 *S. caprae*, and 1 *S. capitis*. The biofilm-producing *S. epidermidis* reference strain ATCC 35984 was also included in the study.

### 4.4. Antimicrobial Assessment

The antimicrobial activity of propolis ethanolic extracts was assessed as described by Queiroga et al. [46], following CLSI protocol M07-A9 [47], in polystyrene flat-bottomed 96-well microtiter plates, in triplicate, by the microdilution methodology for concentrations between 0.05 and 214 mg/mL.

For determining the minimum bactericidal concentration (MBC) of each isolate, 10 μL of each propolis ethanolic extract dilution was inoculated onto a 150 mm diameter Petri dish with Mueller-Hinton Agar using a 96-pin microplate replicator (Boekel Scientific, Feasterville, PA, USA) [46]. The MBC is the lowest dilution of the propolis ethanolic extract able to inactivate the growth of each staphylococcal isolate.

### 4.5. Data Analysis

Since the data followed a normal distribution, as assessed by the Kolmogorov–Smirnov test, an ANOVA was performed to assess the differences in susceptibility of the different *Staphylococcus* species to the propolis ethanolic extract. Significantly different means were determined using Tukey’s honestly significant difference (HSD) test (*p* < 0.05).

### 4.6. Antibiofilm Assessment

Both the inhibitory effect on biofilm formation and the propolis ethanolic extract ability to eliminate established biofilms were assessed, as described by Laranjo et al. [44]. Briefly, bacterial isolates were cultured in flat-bottomed sterile 96-well polystyrene microtiter plates. After incubation, biofilms were stained and measured by optical density assessment at 620 nm using an ELISA plate reader.

For the inhibitory evaluation of staphylococcal biofilm formation, bacterial suspensions were grown with the propolis ethanolic extract half minimum bactericidal concentration for each isolate. The propolis ethanolic extract effect on preformed biofilms was evaluated by adding half MBC of the propolis ethanolic extract on previously grown staphylococcal cultures with established biofilms on 96-well polystyrene flat-bottomed microtiter plates. Mean biofilm inhibition and mean biofilm disruption percentages were calculated by comparison with a biofilm formation assay.

### 4.7. Toxicological Assessment

Canine progenitor epithelial keratinocytes (CPEKs; CELLnTEC, Bern, Switzerland) and human dermal fibroblasts (HDFs; CELL Applications, Inc., San Diego, CA, USA) were cultured in Dulbecco’s modified eagle’s medium (DMEM) (Sigma #D5523, Merck, Germany) supplemented with 10% fetal bovine serum, 1% penicillin/streptomycin, 5 µg/mL insulin, and 10 ng/mL canine epidermal growth factor (cEGF; CPEK) or 10 ng/mL human fibroblast growth factor (hFGF; HDF) and maintained in a 37 °C humidified 5% CO_2_ atmosphere.

Upon reaching 70–90% confluence, cells were detached using 0.25% trypsin–EDTA and collected to be plated. Toxicological studies followed the methodology described by Duran et al. [48]. Briefly, batches of 5000 HDF cells/well or 20,000 CPEK cells/well were plated and allowed to sediment overnight. On the following day, PEE (0.025–2.50 mg/mL) in a culture medium was added. Untreated cells were used to determine the maximum viability (negative control), and 1% Triton was used as a positive control (minimum viability). After 72 h, CCK-8 (Merck, Germany) was added to each well and incubated for 1 h. Afterward, the absorbance (*Abs*) was read at 450 nm (reference at 650 nm), according to the manufacturer’s instructions [49]. Blank controls (without cells) with and without PEE were prepared to evaluate potential interferences. All experiments were performed in triplicate.

The percentage of viability (%V) was determined according to the following mathematical equation:$$\%V=\frac{{Abs}_{\left(Sample\right)}-{Abs}_{\left(Blank\right)}}{{Abs}_{\left(Control\right)}-{Abs}_{\left(Blank\right)}}\times 100$$

## 5. Conclusions

The concentration of 1.0 mg/mL of the propolis ethanolic extract was well-tolerated by dermal fibroblasts and moderately tolerated by epithelial keratinocytes. This dosage was proven to be bactericidal for most staphylococcal isolates. Moreover, it is worth highlighting that both *S. epidermidis* and *S. aureus* isolates, the most important human pathogenic species, are particularly susceptible to propolis. Furthermore, this dosage (1.0 mg/mL) would also be effective in inhibiting the formation of biofilms and disrupting established biofilms.

Our results suggest that propolis is a natural and sustainable alternative to antimicrobials for the control of animal and human infections, namely for topical antibacterial treatment. In this study, we confirmed that a sole propolis ethanolic extract was bactericidal, inhibited biofilm formation, and disrupted pre-formed biofilm, while showing to be moderately to well-tolerated by fibroblasts and epithelial cells. Thus, propolis can be considered a good alternative for multiresistant staphylococcal strains. Further in vivo studies should evaluate propolis as an alternative treatment for infections with *Staphylococcus* spp.

## Figures and Tables

**Figure 1 antibiotics-12-00347-f001:**
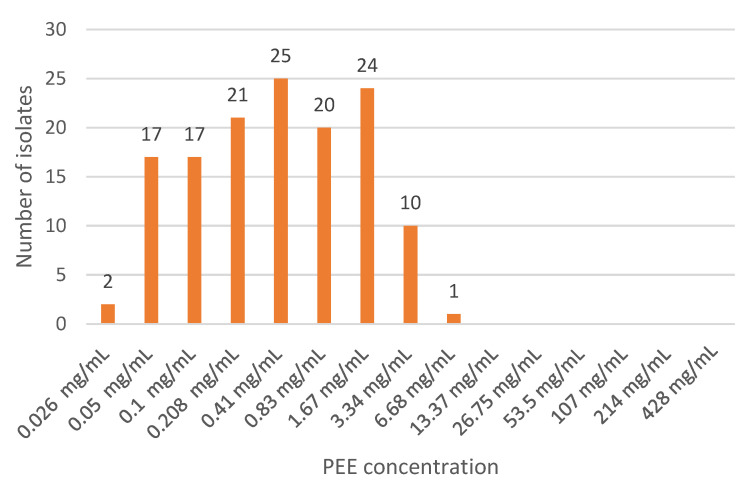
PEE minimum bactericidal concentration against 137 staphylococci field isolates.

**Figure 2 antibiotics-12-00347-f002:**
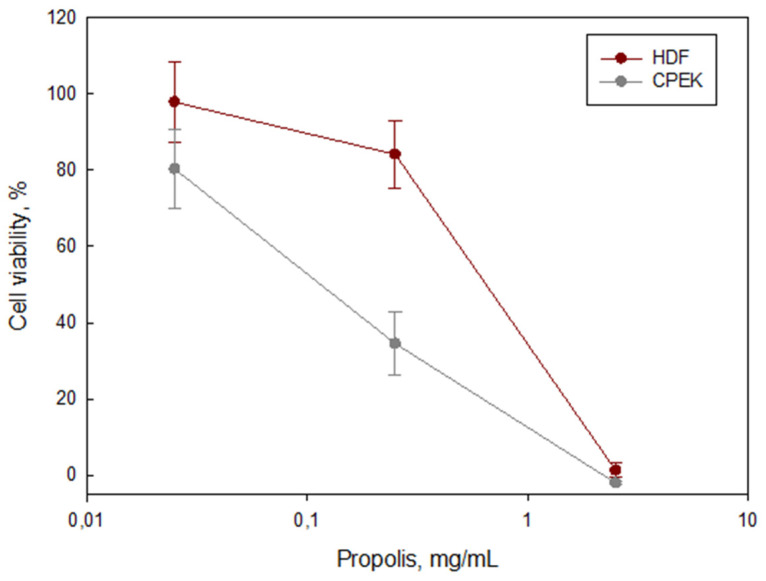
Effect of propolis (PEE) on the cell viability of fibroblasts (HDF) and keratinocytes (CPEK). Control was obtained from untreated cells (viability = 100%); Triton 1% was used as positive control for the loss of cell viability (viability = 0%).

**Table 1 antibiotics-12-00347-t001:** Chemical profile as defined by UPLC-QTOF-MS/MS.

Compound	λ_max_ (nm)	[M-H]^−^ (m/z)	[M-H]^−^ (m/z) Calculated
Pinobanksin	290	271.0662	271.0611
Chrysin	266, 313	253.0756	253.0506
Acacetin	326	283.0817	283.0617
Apigenin	339	269.0714	269.0455
Pinocembrin	289	255.0926	255.0662
Kaempferol-dimethyl-ether	345	313.0936	313.0717

**Table 2 antibiotics-12-00347-t002:** Susceptibility to PEE of the different *Staphylococcus* species.

*Staphylococcus* Species	No. of Isolates	Susceptibility to PEE (mg/mL)

*S. aureus*	40	0.82 ^abc^ ± 0.96
*S. auricularis*	4	0.95 ^abc^ ± 1.59
*S. capitis*	4	1.68 ^abc^ ± 1.34
*S. caprae*	25	0.95 ^abc^ ± 0.81
*S. chromogenes*	19	0.15 ^a^ ± 0.11
*S. epidermidis*	16	0.48 ^ab^ ± 0.55
*S. equorum*	1	0.41 ^abc^ ± 0.00
*S. haemolyticus*	4	0.36 ^abc^ ± 0.31
*S. hominis*	4	0.66 ^abc^ ± 0.76
*S. hyicus*	3	0.05 ^abc^ ± 0.00
*S. lentus*	5	0.84 ^abc^ ± 0.77
*S. rostri*	1	0.83 ^abc^ ± 0.00
*S. simulans*	10	1.47 ^bc^ ± 1.92
*S. warneri*	7	1.98 ^c^ ± 1.40
*Staphylococcus* sp.	1	1.00 ^abc^ ± 0.00

Data are shown as means ± standard deviation. Distinct lowercase letters (a–c) represent significantly different means (*p* < 0.05).

**Table 3 antibiotics-12-00347-t003:** Propolis inhibition of biofilm formation: number and respective percentage of isolates and mean inhibition percentage.

Isolates/Species	*S. aureus*	*S. auricularis*	*S. caprae*	*S. capitis*	*S. chromogenes*	*S. epidermidis*	*S. simulans*	*S. warneri*	Total N/Mean %
N	26	3	1	1	7	1	2	4	45
Inhibited isolates (N)	21	1	1	0	6	1	2	2	34
Inhibited isolates (%)	80.8	33.3	100.0	0.0	85.7	100.0	100.0	50.0	80.7
% of Inhibition	78.4	34.5	61.2	0.0	53.9	63.5	75.9	67.2	71.0

**Table 4 antibiotics-12-00347-t004:** Propolis effect on established biofilms: number and respective percentage of isolates and mean percentage of biofilm disruption.

Isolates/Species	*S. aureus*	*S. auricularis*	*S. caprae*	*S. capitis*	*S. chromogenes*	*S. epidermidis*	*S. simulans*	*S. warneri*	Total N/Mean %
N	26	3	1	1	7	1	2	4	45
Affected isolates (N)	20	2	1	1	6	0	2	4	36
Affected isolates (%)	76.9	66.6	100.0	100.0	85.7	0.0	100.0	100.0	82.9
% of Disruption	88.3	69.2	100.0	76.8	92.5	0.0	100.0	87.4	88.5

## Data Availability

The data presented in this study are available on request from the corresponding author.

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
