# Peer review of "Antimicrobial, Antibiofilm and Toxicological Assessment of Propolis"

_antibiotics, 2023, doi:10.3390/antibiotics12020347_

Round 1

Reviewer 1 Report

The topic turns out to be very interesting. Although there are many works on the antimicrobial properties of propolis, this work focuses on a single genus of bacteria, staphylococcus, and attempts to investigate several aspects of infection, inhibitory/bactericidal activity, biofilm, and toxicity on cells following any treatment.

I think the article is well organized and well written. The discussion could be improved.

Minor comments:

-Page 1 line 40 “vice-versa” should be written in italics.

-Introduction: topics are a bit disconnected, please link paragraphs better.

-pages 1 and 2 lines 45-46, 46-48, 61-63 bibliographic references are missing, please add them.

-page 2 line 63-64 the sentence was interrupted, please finish it.

-page 2 line 63 please write the abbreviation AMR in full at least once.

-page 3 minimum bactericidal concentration has been assessed, but has the minimum inhibitory concentration been investigated? please report the results obtained.

-page 6 lines 167-169 At what concentrations were these substances active? are they comparable to their amounts measured in propolis? please discuss in the text.

-page 6 lines 172-174 bibliographic reference is missing, please add it.

-page 6 lines 175-177 At what concentrations were these substances active? are they comparable to their amounts measured in propolis? please discuss in the text.

-page 6 lines 178-180 At what concentrations was the propolis extract active? Is it comparable to the results obtained by the authors? please discuss in the text.

-page 6 lines 181-185 At what concentrations were these substances active? are they comparable to their amounts measured in propolis? please discuss in the text.

-page 7 lines 225-227 please add some details for each method used.

-page 7 lines 244 what changes were applied? please briefly specify the method used.

-page 7 lines 254-256 please specify how it was measured.

-data analysis: Was a test applied to verify that the data were normally distributed? If yes specify in the text, the test applied otherwise check. If they were not normally distributed, you could not apply the anova test but a nonparametric test should be applied.

Author Response

The topic turns out to be very interesting. Although there are many works on the antimicrobial properties of propolis, this work focuses on a single genus of bacteria, staphylococcus, and attempts to investigate several aspects of infection, inhibitory/bactericidal activity, biofilm, and toxicity on cells following any treatment.

We thank the reviewer for acknowledging our work.

I think the article is well organized and well written. The discussion could be improved.

The discussion and conclusions sections have been improved.

Minor comments:

-Page 1 line 40 “vice-versa” should be written in italics.

Done.

-Introduction: topics are a bit disconnected, please link paragraphs better.

The introduction has been rewritten and new text has been added (page 1, lines 43-44, 47-50, 58, 60, 66-72, 77-78).

-pages 1 and 2 lines 45-46, 46-48, 61-63 bibliographic references are missing, please add them.

New bibliographic references have been included.

-page 2 line 63-64 the sentence was interrupted, please finish it.

This was a mistake. The “comma” has been replaced by a “period”.

-page 2 line 63 please write the abbreviation AMR in full at least once.

The abbreviation of AMR is written in full the first time it appears in the text (page 1, line 38).

-page 3 minimum bactericidal concentration has been assessed, but has the minimum inhibitory concentration been investigated? please report the results obtained.

The minimum inhibitory concentration has not been assessed due to a technical issue, because the propolis ethanolic extract was highly coloured interfering with the OD measurements. However, we will take this into consideration in future studies. One possible solution to overcome this technical issue, may be the determination of the absorbance spectrum of the propolis ethanolic extract, which can then be used to correct the OD measurements.

-page 6 lines 167-169 At what concentrations were these substances active? are they comparable to their amounts measured in propolis? please discuss in the text.

The total content in phenolics, flavonoids, anthocyanins and condensed tannins has been determined in the propolis ethanol extract. The amounts of these groups of compounds in the propolis ethanolic extract are given in the text (page 2, lines 89-91). However, the activity range cannot be given for a group of compounds, it is only given for individual compounds.

-page 6 lines 172-174 bibliographic reference is missing, please add it.

This sentence refers to Table 1.

-page 6 lines 175-177 At what concentrations were these substances active? are they comparable to their amounts measured in propolis? please discuss in the text.

Only the total content in phenolics, flavonoids, anthocyanins and condensed tannins has been determined in the propolis ethanol extract. We did not quantify individual compounds.

-page 6 lines 178-180 At what concentrations was the propolis extract active? Is it comparable to the results obtained by the authors? please discuss in the text.

To determine both the biofilm inhibition and biofilm disruption abilities, the half minimum bactericidal concentration of the propolis ethanol extract for each staphylococcal isolate was used, as explained in the Materials and Methods section (page 8, lines 286-292).

-page 6 lines 181-185 At what concentrations were these substances active? are they comparable to their amounts measured in propolis? please discuss in the text.

Only the total content in phenolics, flavonoids, anthocyanins and condensed tannins has been determined in the propolis ethanol extract. Individual compounds were not quantified.

-page 7 lines 225-227 please add some details for each method used.

Details have been added in the text (page 7, lines 236-239).

-page 7 lines 244 what changes were applied? please briefly specify the method used.

The used protocol is fully detailed in Laranjo, et al. [1].

-page 7 lines 254-256 please specify how it was measured.

A sentence has been added to the text to better explain this (page 7, lines 269-270).

-data analysis: Was a test applied to verify that the data were normally distributed? If yes specify in the text, the test applied otherwise check. If they were not normally distributed, you could not apply the anova test but a nonparametric test should be applied.

Yes, the data are normally distributed, as asses with the Kolmogorov-Smirnov test. This information has been added to the manuscript (page 7, lines 274-275).

  1. Laranjo, M.; Andrade, N.; Queiroga, M.C. Antibiofilm activity of propolis extracts. In Understanding microbial pathogens: current knowledge and educational ideas on antimicrobial research, Méndez-Vilas, A., Ed. Formatex Research Center: Badajoz, 2018.

Reviewer 2 Report

I recomend this manuscript to be published in Antibiotics as could be useful for some research in propolis.

Please in table 2, make more clear the significance of abc

Author Response

I recomend this manuscript to be published in Antibiotics as could be useful for some research in propolis.

We thank the reviewer for acknowledging our work.

Please in table 2, make more clear the significance of abc

Distinct lowercase letters (a–c) represent significantly different means (p < 0.05) according to the ANOVA post-hoc Tukey’s HSD test (page 7, line 276-278).

Reviewer 3 Report

I read the paper with interest, as any alternative to conventional antibiotics deserves attention. It is an interestingly written methodological study. However, I’d like to know  what is the novel aspect of this study.

Please indicate the novelty of this work more clearly as the antimicrobial properties of propolis are well known.

Minor comments;

Line 64; please add the reference to this statement

Lines 308-311; This is well known statement. What’s new here?

Author Response

I read the paper with interest, as any alternative to conventional antibiotics deserves attention. It is an interestingly written methodological study. However, I’d like to know what is the novel aspect of this study.

We thank the reviewer for the most valuable comments.

Please indicate the novelty of this work more clearly as the antimicrobial properties of propolis are well known.

The novelty of the current study is the fact that a sole propolis ethanol extract was bactericidal, inhibited biofilm formation and disrupted pre-formed biofilm, while showing to be moderately to well tolerated by both fibroblasts and epithelial cells.

Minor comments:

Line 64; please add the reference to this statement

A new bibliographic reference has been included.

Lines 308-311; This is well known statement. What’s new here?

The novelty of the current study is the fact that a sole propolis ethanol extract was bactericidal, inhibited biofilm formation and disrupted pre-formed biofilm, while showing to be moderately to well tolerated by both fibroblasts and epithelial cells. A sentence explaining the novelty of our study has been included in the conclusions section (pages 9, lines 326-329).

Round 2

Reviewer 3 Report

I accept the manuscript in present form and recommend it for publication.